# Manufacture and Characterization of Alginate-CMC-Dextran Hybrid Double Layer Superabsorbent Scaffolds

**Jeongyeon Choi** [1,*,†] **and Heekyung Jeon** [2,†]

1  Advanced Mechatronics R&D Group, Daegyeong Division,
   Korea Institute of Industrial Technology (KITECH), 320 Technosunhwan-ro, Yuga-eup,
   Dalseong-gun, Daegu 42994, Korea
2  Advanced Energy Materials and Components R&D Group, Dongnam Division,
   Korea Institute of Industrial Technology (KITECH), 804 Baekyang-daero, Sasang-gu, Busan 46938, Korea;
   jeonhk75@kitech.re.kr
*  Correspondence: jychoi77@kitech.re.kr; Tel.: +82-53-580-0194; Fax: +82-53-580-0120
†  These authors contributed equally to this work.

**Abstract:** This study focused on the manufacturing of functional superabsorbent sponges using natural polymers. An alginate/CMC-embedded dextran hybrid dual-layer formulation was prepared using the freeze-drying method. The physical properties of the formulation were characterized using a field emission scanning electron microscope and a universal testing machine, and the swelling ratio was calculated. Cell viability assays were performed using keratinocytes (HaCaT cells). The results showed that this formulation can absorb a large amount of moisture and provide morphological stability through its tensile strength and uniform porosity, and this was verified by its biocompatibility. We believe that in the future, by combining this novel hybrid dual-layer superabsorbent sponge with antibacterial agents with excellent porosity, it would serve as a medical material for producing bandages that can absorb blood and body fluids, feminine hygiene products, and functional antibacterial masks.

**Keywords:** superabsorbent; medical materials; CMC; alginate; dextran

## 1. Introduction

Recently, the demand for health care and sanitary textile products increased. These products are required for a hygienic, clean, and comfortable life due to the effects of COVID-19, as well as for promoting better health conditions and preventing problems caused by environmental pollution, such as with fine dust. Interest in preventive measures such as the use of masks, personal hygiene products, and disposable hygiene products is thus increasing. Currently, instead of producing sanitary products using non-woven fabrics, material studies are actively underway on the use of superabsorbent materials that can contain a large amount of water, without releasing absorbed water even under pressure. A typical scaffold form of a superabsorbent material is a hydrogel sponge. Hydrogels have a three-dimensional cross-connecting network structure in which the hydrophilic polymer is not dissolved in solvents and may contain a large amount of water or liquid (up to 1000 times or more of the weight of the polymer) [1–3].

Commercially widely used superabsorbent sponges are polymerized by adding polymerization initiators and crosslinking agents based on polyvinyl chloride (PVC), acrylic acid extracted from petroleum, and acrylamide. Currently, there are developments on superabsorbent sponges, but most of them are used as heavy metal adsorbents, soil moisture control agents for agriculture or horticulture, food packaging materials, and construction materials. However, its application to medical materials (diapers for infants and adults, sanitary napkins for women) and disposable sanitary products is low due to concerns about toxicity and environmental pollution [4,5].

Therefore, the most important purpose of this study was to develop new ultra-highly absorbent medical materials using natural polymers with non-toxicity, biocompatibility,

biodegradability, wide specific surface area, and abundant reactivity. Natural polymers, including plant-derived, animal-derived, and microbial-producing polymers, are composed of polysaccharides such as carbohydrates, proteins, and cellulose, and thus have biocompatibility and biodegradability characteristics. Examples of natural polymers include alginate (Alg), dextran (Dex), and carboxymethyl cellulose (CMC). Alginate is widely used to manufacture various types of superabsorbent scaffolds because it can be made into a product with excellent functions owing to its copolymer structure in which a mannuronic acid (M) block, guluronic acid (G) block, and intermediate M/G block are linked by 1,4 glycosides. Since there is a carboxyl group (COOH-) in the alginate molecule, it is acidic and generally has a hydrogel formulation by ionic bonding with sodium salt (Na+) to form an egg model. Alginate in the form of its sodium salt may be developed into various types of hydrogels, capsules, or sponges through cross-linking with metal ions ($Ca^{2+}$, $Ba^{2+}$). In particular, alginate is a water-soluble natural polymer polysaccharide with a negative charge and exhibits excellent antibacterial effects upon ionic bonding, with biodegradability, high water content, and non-toxicity [6,7]. Dextran is highly water soluble and biocompatible, and is widely applied in biomedical medicine. Dextran is a complex and high molecular weight polysaccharide composed of glucose molecules made by bacterial decomposition of sugar cane candy, and the main chain is composed of $\alpha$ (1→6) glycosidic bonds between glucose monomers. The branched portion is connected by $\alpha$ (1→4), $\alpha$ (1→2) or $\alpha$ (1→3) glycosidic bonds. This characteristic branching distinguishes dextran and dextrin, and physicochemical properties appear differently, depending on the number of branched links and glucopyranosyl residues. Owing to these properties, dextran is widely used for hydrogel formulations and drug delivery systems. In particular, dextran is widely used as an antithrombotic agent that can be applied as a dressing and bandage, and it is effective as a coating agent that protects and stabilizes metal ions from oxidation [8,9]. Hydrophilic cellulose (CMC), a natural biomass resource widely used as a superabsorbent material, is a biodegradable natural polymer produced by microorganisms, and is used in various cosmetics, disposable diapers, and women's hygiene products (sanitary pads) because it is stable for skin moisturization and has ultra-absorbent properties. However, pure cellulose that is not chemically modified has a reduced ability to absorb and retain moisture. This limits its application in disposable sanitary products. However, it has the advantage of increasing mechanical strength and controlling the decomposition period by fusion with biomaterials [10–12].

Therefore, in this study, by using natural polymers with eco-friendly and biocompatibility functions, we developed highly absorbent materials and attempted to ascertain their potential utility as medical materials. In addition, we considered the idea of a double structure with excellent moisture absorption and tensile strength, and devised a sponge formulation with a double structure (Figure 1).

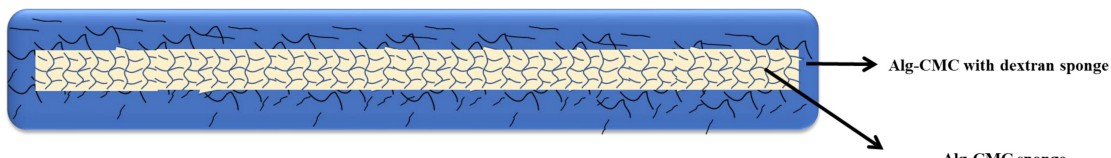

**Figure 1.** Concept of Alg-CMC hybrid dextran double-layer superabsorbent sponges (ACDSs).

## 2. Materials and Methods

### 2.1. Manufacturing Method

ACDS was prepared using a freeze-drying method [13,14]. Briefly, 5.0 wt% CMC (carboxymethyl cellulose, viscosity, mPa·s 1.0 %s resolution 25,000, Sigma-Aldrich, St. Louis, MO, USA) was added to 1.0 wt% sodium alginate (LVG, viscosity: 20–200 mPa·s, NovaMatrix, Sandvika, Norway) and uniformly mixed. They were then lyophilized for 48 h in a freeze dryer (DFType, Ilshin Lab. Co., Ltd., Yangju-si, Korea) at −50 °C to −60 °C to prepare the Alg-CMC. Next, 1.0 wt% calcium chloride dehydrate ($CaCl_2 \cdot 2H_2O$, Sigma,

St. Louis, MO, USA) was left at room temperature for 24 h to cause crosslinking, and then a calcium solution was completely removed using triple-distilled water (DW, Milli-Q®, Waltham, MA, USA) and 95% ethyl alcohol to produce the Alg-CMC sponge. Thereafter, 10.0 wt% dextran (Sigma-Aldrich, St. Louis, MO, USA) was mixed with 1.0 wt% NaOH (Sigma-Aldrich, St. Louis, MO, USA) and stirred at room temperature at a low speed (80 rpm), and then mixed with a dextran solution. Then, 0.5 wt% BDDE (1,4-butanediol diglycidyl ether (Sigma-Aldrich, St. Louis, MO, USA) was immersed in the mixture and crosslinked in a 50 °C water bath for 4 h. The crosslinking agent was washed three to five times for 24 h using triple distilled water (DW, Milli-Q®, Waltham, MA, USA) to remove unreacted materials. It was then lyophilized for 48 h in a lyophilization chamber at −50 °C to −60 °C to prepare the Alg-CMC hybrid dextran double-layer superabsorbent sponges (ACDSs) (Table 1, Figure 2).

**Table 1.** Processing conditions for fabrication of Alg-CMC hybrid dextran double-layer superabsorbent sponges (ACDSs).

| Samples | Type of Sponges | Embedded Dextran |
|---|---|---|
| Control | Only Alg | 0 |
| a | Alg-CMC (ratio) 5.0:5.0 | 0 |
| b | Alg-CMC (ratio) 5.0:5.0 | 5.0 |
| c | Alg-CMC (ratio) 7.0:3.0 | 0 |
| d | Alg-CMC (ratio) 7.0:3.0 | 5.0 |

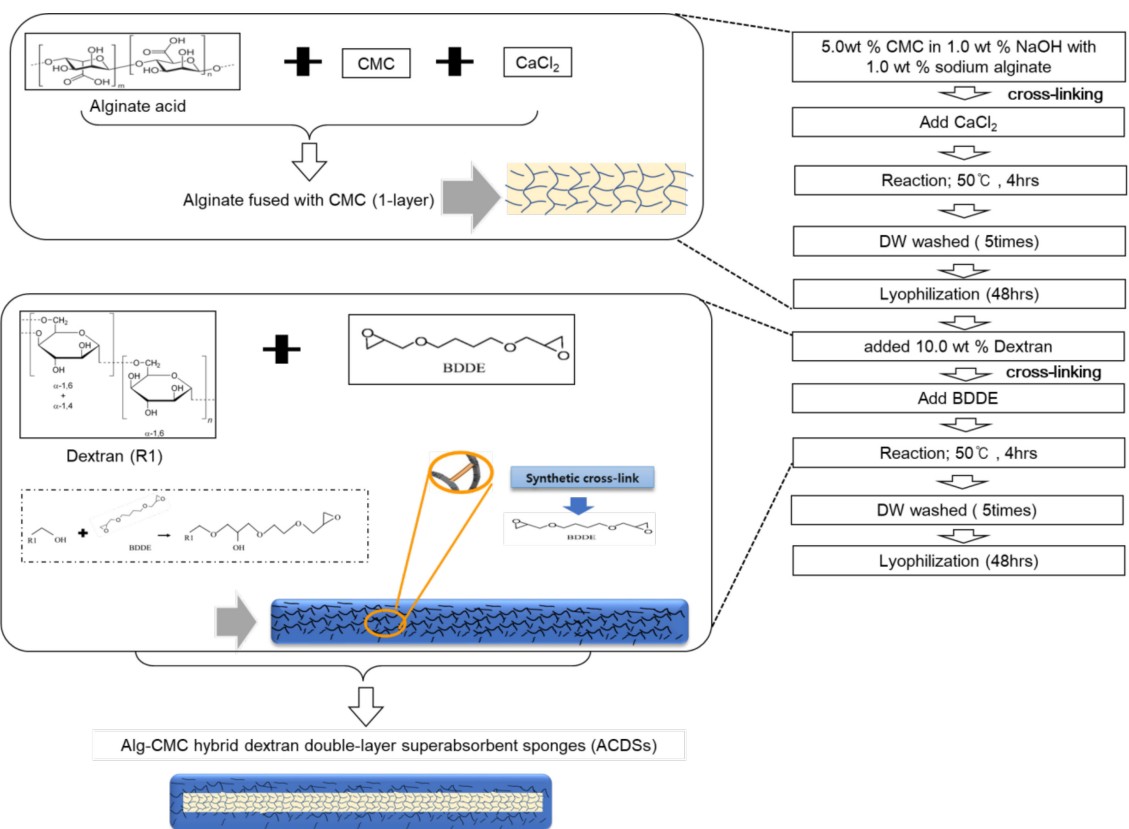

**Figure 2.** Schematic diagram showing the fabrication process of the Alg-CMC hybrid dextran double-layer superabsorbent sponges (ACDSs).

## 2.2. Evaluation of Morphological and Mechanical Properties

The surface morphology of the prepared ACDSs was analyzed using a field emission scanning electron microscope (FE-SEM) (Hitachi S-4300 and EDX-350, Hitachi, Chiyoda City, Japan). The tensile strength of the prepared ACDSs was measured using an Instron 5965 universal testing machine (UTM).

## 2.3. Analysis of the Gelation Rate and Swelling Ratio

To measure the gelation rate of the ACDSs, the initial weight (Wi) of the prepared patch was measured. Then, the ACDS was immersed in 50 mL of ultrapure water and stirred in a shaking water bath (100 rpm) at 37 °C for 12 h. The ACDS was then placed in a freeze dryer. The dried weight (Wd) was measured, and the gelation rate was calculated using Equation (1) [15]:

$$\text{Gelation rate (\%)} = \text{Wd}/\text{Wi} \times 100. \tag{1}$$

To calculate the swelling ratio, the initial dried weight (Wd) was measured after drying the patch at room temperature. The ACDS was then immersed in 50 mL ultrapure water and placed in a shaking water bath (100 rpm) at 37 °C for 24 h (Ws, swelling weight) until the absorption equilibrium of the ACDS was reached, and the swelling ratio was calculated using Equation (2) [15]:

$$\text{Swelling ratio (\%)} = (\text{Ws} - \text{Wd})/\text{Wd} \times 100. \tag{2}$$

## 2.4. Cell Compatibility Analysis

The ACDS was disinfected by immersing it in 70% EtOH under UV light for 1 d. The ACDS was then washed several times with triple-distilled water before use. Next, the disinfected ACDS was immersed in a cell culture medium consisting of Dulbecco's modified Eagle's medium (DMEM, Gibco, Carlsbad, CA, USA) containing 10% fetal bovine serum, 500 U/mL penicillin, and 500 μg/mL streptomycin (Gibco, Carlsbad, CA, USA). After incubating the patch for 2 h at 37 °C and 5% $CO_2$, $1 \times 10^5$ human keratinocyte cell line (HaCaT) cells were incubated for 24 h (37 °C, 5% $CO_2$). Then, 3-(4,5-dimethylthiazol-2-yl)-5-(3-carboxymethoxyphenyl)-2-(4-sulfophenyl)-2H-tetrazolium, inner salt (MTS, Promega, Madison, WI, USA) solution was added to the eluted medium for 4 h (37 °C, 5% $CO_2$). Absorbance at 490 nm was measured using an ELISA reader (Infinite® 200 PRO, Tecan Trading AG, Männedorf, Switzerland) to evaluate the cell viability ($n$ = 3).

## 2.5. Statistical Analysis

Data are expressed as mean ± standard deviation (SD). The collected data were compared using *t*-tests, and the different-types of data were evaluated using Microsoft Excel 2010 (Microsoft, Redmond, WA, USA).

## 3. Results

### 3.1. Morphological and Mechanical Analysis of Scaffolds

From the visual observation of the prepared scaffolds with a digital camera, a smooth surface was observed for the control alginate sponge, and a uniform surface was observed for the sponge containing CMC in alginate. In addition, it was confirmed that the dextran-treated hybrid two-layer sponge was molded on a plate of the same diameter, but expanded in volume compared to the control, and it had a smooth and uniform shape. This indirectly demonstrates that dextran can maintain stable moisture through excellent absorbency and crosslinking (Figure 3).

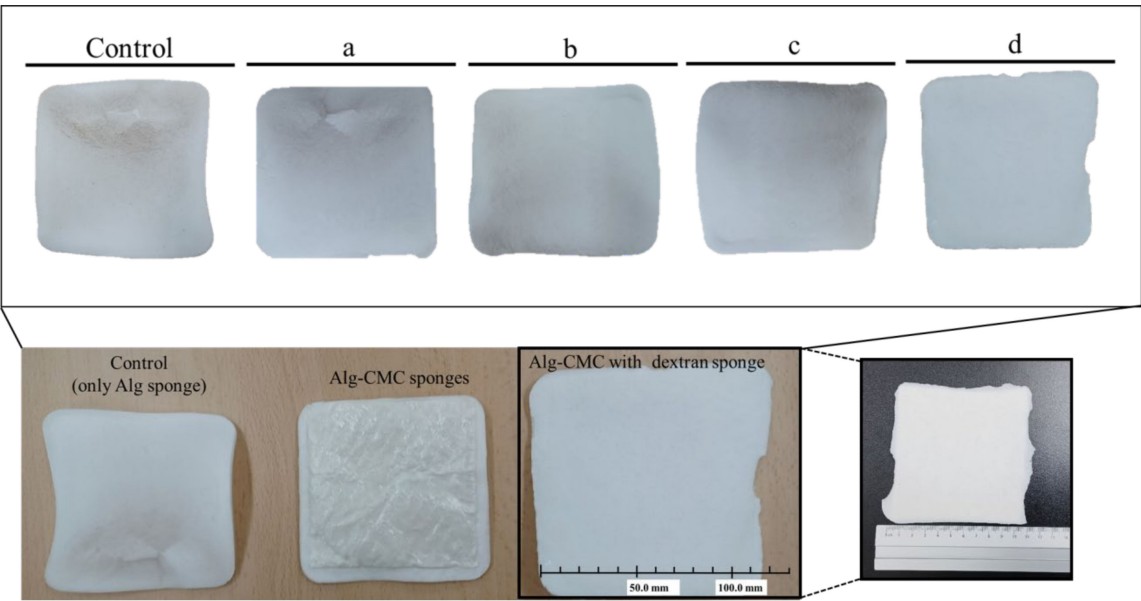

**Figure 3.** Image of a control sponge; (**a**) Alg-CMC ratio (5:5); (**b**) Alg-CMC ratio (5:5) with dextran; (**c**) Alg-CMC ratio (7:3); (**d**) Alg-CMC ratio (7:3) with dextran (taken by Digital Camera Samsung).

In addition, from observing the surface using the FE-SEM, the formation of pores on the surface increased as the CMC concentration increased, and the pore size (>1 mm) confirmed that the structure had good interpore connectivity (Figure 4a,c). This suggests a phenomenon in which hydrophilic CMC has excellent water content but cannot be completely distributed because it flows into an aqueous solution with high viscosity [16].

In the case of the two-layer sponge using dextran, the pore size (approximately 500 μm or more) was irregular, and the shape of the pores was distorted compared to the control group (Figure 4b,d).

It was thus confirmed that the sponge with high CMC content had a distorted and irregular shape in the double hybrid sponges, and the results show that the high-viscosity aqueous solution CMC was not completely distributed [16].

The tensile strength was measured using UTM to measure the strength of the wet state and the strength of the dry state. Compared with the control group, the one-layer sponge group (Figure 5a,c) showed an increase in the number of holes and a large surface area, which was measured and found to be low in tensile strength due to a decrease in bonding strength caused by pore–pore interaction. In addition, the two-layer sponge group (Figure 5b,d) had a higher tensile strength than the one-layer sponge group (Figure 5a,c). It seems that the tensile strength increases as dextran has a stable structure through cross-linking.

### 3.2. Evaluation of Gelation Rate and Swelling Ratio

In this experiment, the gelation rate and swelling behavior of the hybrid two-layer sponge material were evaluated. Figure 6a,c having a single structure with CMC in the wet state compared to the dry state of the same size, showed an increase in the moisture content compared to the control group; in particular, it was visually evaluated that a large amount of moisture was absorbed in the double structure hybrid with dextran (Figure 6b,d).

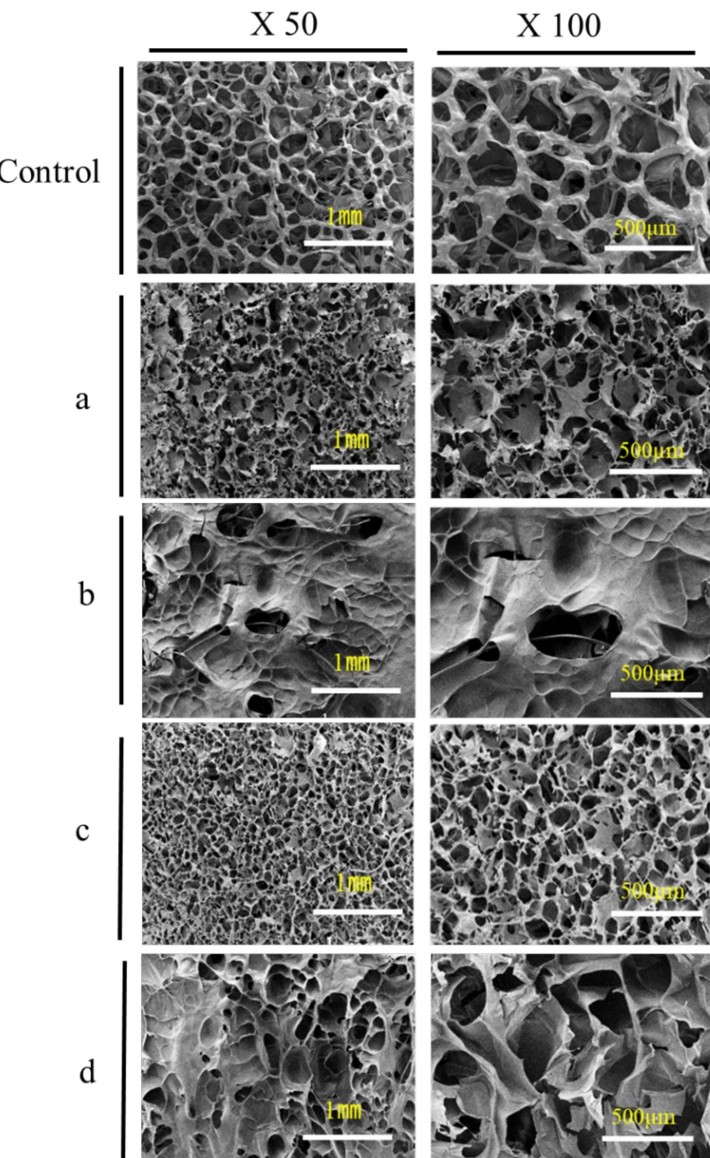

**Figure 4.** Morphological characterization with field emission scanning electron microscope (FE-SEM); control sponge, (**a**) Alg-CMC ratio (5:5); (**b**) Alg-CMC ratio (5:5) with dextran; (**c**) Alg-CMC ratio (7:3); (**d**) Alg-CMC ratio (7:3) with dextran (scale bars 5.0 kV $\times$ 5.0 K).

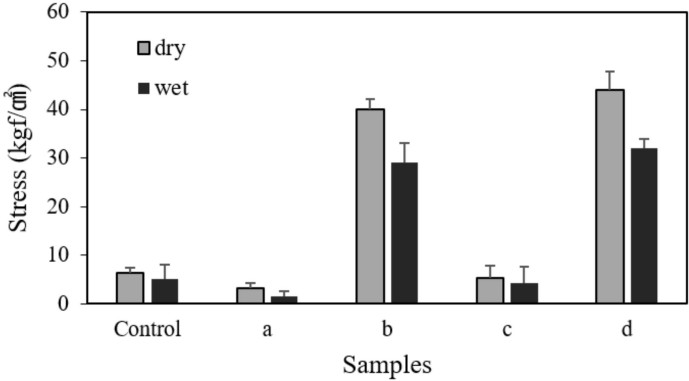

**Figure 5.** Tensile strength of control sponge, (**a**) Alg-CMC ratio (5:5); (**b**) Alg-CMC ratio (5:5) with dextran; (**c**) Alg-CMC ratio (7:3); (**d**) Alg-CMC ratio (7:3) with dextran ($n = 3$, mean $\pm$ SD).

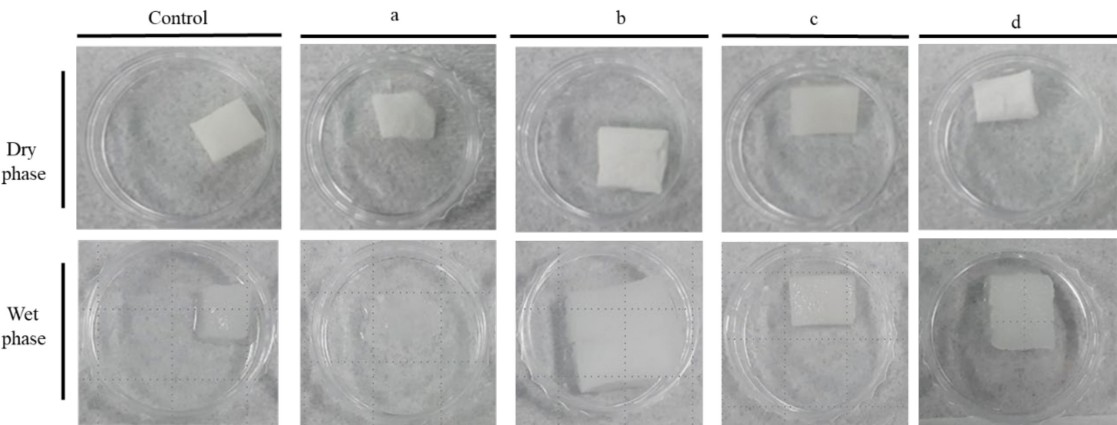

**Figure 6.** Photographs of subsequent lyophilization (dry phase, **top**) and water absorption (wet phase, **bottom**); control sponge, (**a**) Alg-CMC ratio (5:5); (**b**) Alg-CMC ratio (5:5) with dextran; (**c**) Alg-CMC ratio (7:3); (**d**) Alg-CMC ratio (7:3) with dextran.

It was confirmed through the gelation rate that the water absorption was slightly increased in the sponge with a high CMC ratio (Figure 7Aa,c). There was a high gelation rate of 70% or more in all the groups, and in particular, the group containing dextran showed a tendency to increase (Figure 7Ab,d). The swelling behavior was observed for 24 h, and as a result, it was confirmed that the swelling property increased in the dextran-containing group (Figure 7Bb,d). In general, dextran is a hydrophilic polymer and it increases the absorption of water.

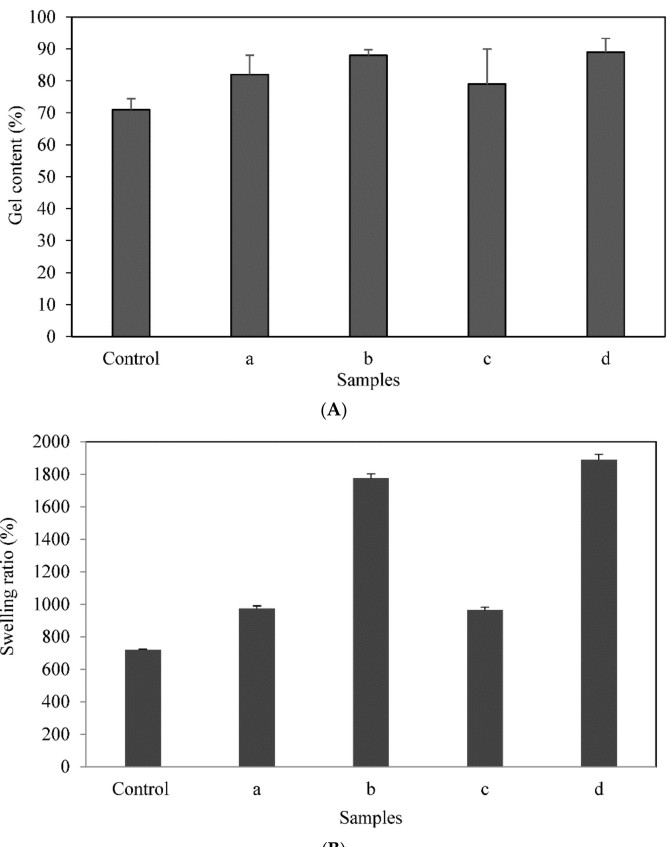

**Figure 7.** Effect of gel content (**A**) and swelling ratio (**B**); control sponge, (**a**) Alg-CMC ratio (5:5); (**b**) Alg-CMC ratio (5:5) with dextran; (**c**) Alg-CMC ratio (7:3); (**d**) Alg-CMC ratio (7:3) with dextran ($n$ = 3, mean ± SD).

### 3.3. Biocompatibility Evaluation

The cell viability was evaluated over 4, 24, 48, and 72 h by obtaining a medium containing HaCaT cells. The proliferation rate of cells over time showed a similar tendency (Figure 8). With respect to cell proliferation up to 72 h, it was confirmed that the cell proliferation rate was slightly increased in the group containing dextran, and the results showed the most effective biocompatibility in the Alg-CMC ratio (5:5) with the dextran group (Figure 8b). The higher the content of CMC, greater is the potential for the material to have characteristics of non-toxicity, biocompatibility, biodegradability, wide non-specific surface, and rich reactant group [17].

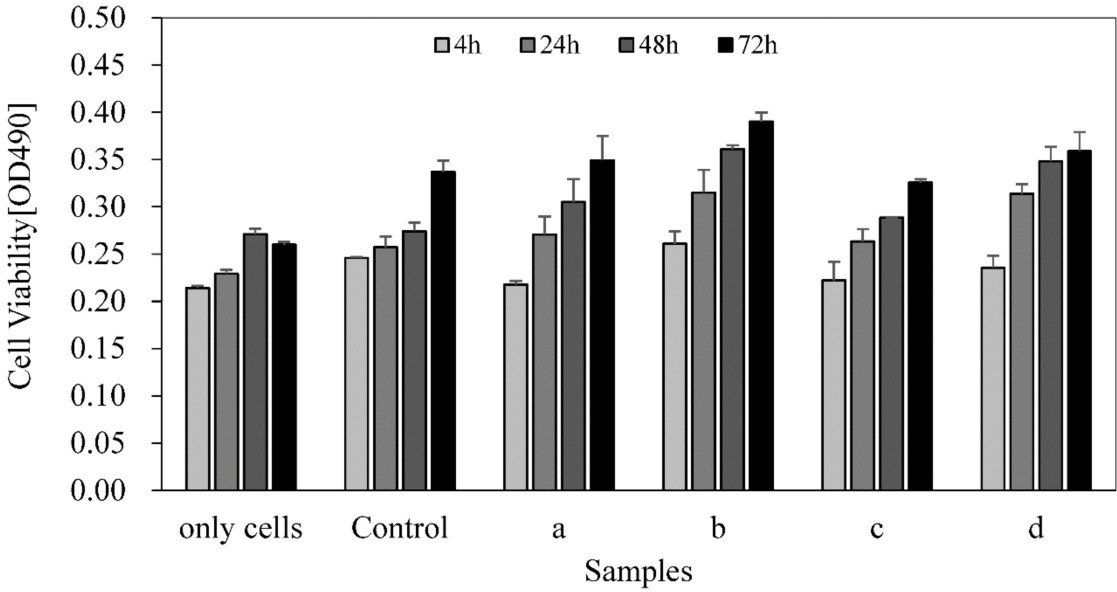

**Figure 8.** Cell viability was tested after obtaining the medium eluted from the (human keratinocyte cell line, HaCaT cells); only cells, control sponge, (**a**) Alg-CMC ratio (5:5); (**b**) Alg-CMC ratio (5:5) with dextran; (**c**) Alg-CMC ratio (7:3); (**d**) Alg-CMC ratio (7:3) with dextran (MTS assay, *n* = 3, mean ± SD).

### 4. Discussion

In this study, to develop superabsorbent medical materials, a scaffold with a hybrid double-layer structure was manufactured using natural polymers, and the physical properties and cell viability were evaluated. Alginate and CMC were mixed and reacted with a cross-linking agent, and then a sponge with a single structure (one-layer) was prepared through a freeze-drying method, which was reacted with a BDDE cross-linking agent, and then a double-structure (two-layer) sponge was prepared through the same process. In Alg-CMC with a single structure (one-layer), the lower the CMC content, the higher the number of holes, and the uniform porosity and slightly increased tensile strength confirm the stability of the material. In all the groups, the tensile strength in the dry state was measured and found to be higher than that in the wet state. The sponge hybrid with dextran (a dual structure) indirectly shows that the tensile strength in a dry state is approximately 40 to 44 N and the tensile strength in a wet state is approximately 30 to 32 N, which can thus absorb a large amount of moisture and blood and may be used as a material with stable mechanical strength. In addition, it was confirmed that the survival rate of human keratinocytes was improved, and the stability of the material was confirmed through cell compatibility. In conclusion, it was confirmed that the developed hybrid double superabsorbent sponge material can contain and maintain mechanical properties and a large amount of moisture. This material is functional, biocompatible, and cell effective. In the future, it is expected to be used in a variety of ultra-absorbent medical materials, such as for wound treatment (bandages, wound bands) and drug delivery systems with intelligent hydration gel, tissue engineering scaffolds, and moisture sensors.

**Author Contributions:** Conceived and designed the experiments, J.C. and H.J.; performed the experiments, J.C.; analyzed the data, J.C. and H.J.; contributed reagents/materials/analysis tools, H.J.; wrote the paper, J.C. All authors have read and agreed to the published version of the manuscript.

**Funding:** This research was funded by the Korea Institute of Industrial Technology (JA210007, development of the green-hydrogen production system by alkaline-electrolysis/desalination and core parts).

**Institutional Review Board Statement:** Not applicable.

**Informed Consent Statement:** Not applicable.

**Data Availability Statement:** Not applicable.

**Acknowledgments:** This study was supported by the Korea Institute of Industrial Technology (JA210007, development of the green-hydrogen production system by alkaline-electrolysis/desalination and core parts).

**Conflicts of Interest:** The authors declare no conflict of interest.

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
