# Peer review of "Manufacture and Characterization of Alginate-CMC-Dextran Hybrid Double Layer Superabsorbent Scaffolds"

_applsci, doi:10.3390/app112311573_

Round 1

Reviewer 1 Report

This paper shows results on manufacturing of functional superabsorbent sponges using natural polymers and characterization of its physical properties.

Here are some suggestions for improvement of the paper:

  1. The abreviation ACDS is used for "Alg-CMC hybrid dextran double-layer superabsorbent sponges". Additionally, HDSS is introduced in line 91 for "Hybrid Dextran double-layer Superabsorbent Sponges", this additional abreviation seems unneccessary, it is not used in the text.
  2. Section 2.5. Statistical analysis contains basic information, which is not specific, thus appears to be retundant.
  3. Figure 6 shows photographs of subsequent lyophilization of samples - discussion of those results should also be included in the text.

Author Response

Dear Reviewer

Thank you for your thorough review of our paper, entitled “Manufacture and Characterization of Alginate-CMC-Dextran Hybrid Double Layer Superabsorbent Scaffolds”.

Best regards, 

Jeong Yeon Choi*, Hee Kyung Jeon

Reviewer 2 Report

  1. In the abstract, I recommend removing the section that refers to the usefulness of hygiene products. The abstract must include only this information: the purpose of the paper, the methods of analysis or characterization used and the results obtained.
  2. The sentence between lines 39-41 must be corrected. Also, please note that the definition of hydrogels was not given for the first time by: Gawande, Capanema or Behrouzi, for example.
  3. Also, the meaning of the sentence between lines 59-60 must be corrected.
  4. In the introduction, what is the purpose of the paper? Please detail the purpose of the paper and what is new in this study compared to current developments.
  5. As the introduction is made, the current state of developments in the field is not explicitly highlighted. I belive the introduction should include information on the development of "superabsorbent sponge" materials, not just general information about the polymers from which this material is obtained.
  6. The section of materials and methods must be somehow separated and presented distinctly. Which was the final pH of the polymer blends before crosslinking.
  7. Rows 103-104: please correct the sentence “The crosslinking agent was then completely removed “by washing” five times with triple DW”.
  8. 2, the crosslinking appears after the addition of CaCl2, respectively BDDE, it does not seem to me represented correctly in the figure in the article.
  9. Figures 3 and 4 should be increase the font of the sample name and image magnification.
  10. What is the reason why the article does not have a section dedicated to the conclusions?
  11. 7 in the explanations, the two graphs must be differentiated, but not with “top” and “bottom”, but for example 7A and 7B. "Bottom" refers to the figure at the bottom, but in fact the figure appears at the top of page 8.
  12. Please detail in figure 6 caption the meaning of a,b,c, d?

Author Response

(The authors gave the same response as above.)
